# Carbon information disclosure and corporate financial performance—Empirical evidence based on heavily polluting industries in China

**Ailing Xu** [ORCID]ᵒ, **Yuanyuan Su** [ORCID]ᵒ, **Yingxin Wang, Jia Liao** [ORCID]*

Business School, Huaqiao University, Quanzhou, Fujian, China

ᵒ These authors contributed equally to this work.
* 20002@hqu.edu.cn

**Data Availability Statement:** All relevant data are within the manuscript and its Supporting Information files.

## Abstract

Global climate change has become one of the most large-scale, widespread, and far-reaching challenges facing mankind. Against this background, China has proposed a "dual-carbon" target in 2020, which greatly demonstrates China's determination and commitment to carbon emission reduction, and the burden of realizing the "dual-carbon" target is mainly borne by heavy polluters. The burden of achieving the "dual-carbon" goal is mainly borne by the heavily polluting firms. Although this has increased the economic burden of the firms to a certain extent, carbon information disclosure reduces the degree of information asymmetry and also obtains the support of the government, which improves the financial performance of the firms. Based on the data of A-share listed companies in Shanghai and Shenzhen in the heavy pollution industry in 2013–2023, this paper analyzes the relationship between carbon information disclosure, and corporate financial performance according to signaling theory, rent-seeking theory, and sustainable development theory. It is found that enhanced corporate carbon disclosure can significantly improve corporate financial performance, and the main effect is realized through reducing debt financing costs and increasing the proportion of institutional investors' shareholding. In the heterogeneity analysis, this paper finds that the main effect is more significant in the samples of firms located in the western region and the central region. Based on existing research, this paper deepens the study of the relationship between carbon disclosure and corporate financial performance. By integrating the multiple perspectives of signaling theory, rent-seeking theory and sustainable development theory, this paper systematically analyzes how creditors, institutional investors and other stakeholders play a role in the dynamic interaction between carbon disclosure and corporate financial performance, and reveals the motives and mechanisms behind the behaviors of these stakeholders. In order to further refine the analysis path, this paper constructs an intermediary model in order to deconstruct the deep logic of the path and mechanism through which carbon disclosure indirectly affects corporate financial performance. This model not only enhances the theoretical explanatory power, but also provides a more refined analytical framework for empirical testing, which helps to reveal the "black box" mechanism of carbon disclosure's impact on corporate financial performance. In addition, in view of China's vast territory and the uneven level of economic development between

**Funding:** Fujian Provincial Social Science Foundation Project, grant numbers: FJ2022B082. The funders had no role in study design, data collection and analysis, decision to publish, or preparation of the manuscript.

**Competing interests:** The authors have declared that no competing interests exist.

regions, this paper adopts a differentiated analysis strategy, based on the economic characteristics of the regions where the heavy polluters are located, and divides the whole sample into three sub-samples for independent regression analysis. This heterogeneity test incorporates inter-regional development differences into the scope of analysis, making the research conclusions more geographically specific and policy-guiding significance. By comparing and analyzing the differences in the impact of carbon disclosure on the financial performance of enterprises in different regions, this paper provides a reference for the government to formulate differentiated carbon disclosure policies in the future, accurately promote the construction of the carbon trading market, and efficiently achieve the carbon emission reduction targets at the national level.

## 1. Introduction

Against the background of global commitment to combating climate change and promoting carbon emission reduction, corporate carbon disclosure has gradually become a hot topic for academic discussion and constitutes a key part of the active fulfillment of carbon emission reduction commitments by countries around the world. Corporate carbon disclosure is an important way for stakeholders to understand the environmental dynamics of corporations and accelerates the greening process of the capital market. In the nascent stage of carbon disclosure, firms are faced with the trade-off between environmental and economic benefits [1], and many of them are cautious about this, with a low willingness to disclose, resulting in limited breadth and depth of information disclosure. In recent years, with the spread of the low-carbon concept, and the increasing attention of the community to environmental issues, investors' environmental preferences have increased significantly, and they are more inclined to invest in firms that actively disclose carbon information [2]. This shift in market demand has prompted firms to re-examine the value of carbon information disclosure, and have increased their investment to improve the quality and level of disclosure, which has gradually led to the standardization and popularization of carbon information disclosure. However, the degree of carbon information disclosure in China is still low, mainly due to the lack of motivation of firms to disclose and the lack of corresponding disclosure standards, and firms are unwilling to spend more costs to disclose carbon information that is only a formality [3]. Moreover, there is a large gap in carbon information disclosure among industries in China, with various forms of disclosure and inconsistencies in the quantity and quality of disclosure [4]. When discussing the maturity and completeness of the global carbon disclosure system, the EU and the UK have undoubtedly demonstrated a more advanced and systematic system structure. Compared with other countries and regions, the two systems in the field of carbon information disclosure are constructed in detail, setting a benchmark for the development of carbon information disclosure system in the international arena. Specifically, the carbon disclosure system implemented by the European Union, through a series of strict norms and guidelines, explicitly requires enterprises to incorporate into their disclosure reports strategic considerations, the construction of policy implementation frameworks, as well as quantitative assessment of the effectiveness of specific emission reduction actions, which greatly enhances the transparency of the disclosure of information and comparability [5]. In addition, the U.S. Securities and Exchange Commission (SEC) has responded positively to the challenge of global climate change in recent years by promulgating new climate change disclosure rules in 2022. These rules not only emphasize the need for companies to follow internationally accepted

standards for reporting greenhouse gas emissions, but also explicitly require companies to submit certified data to ensure the accuracy of the information. This marks a significant advancement in carbon disclosure regulation in the United States, and further promotes the harmonization and enhancement of global carbon disclosure standards. At the level of academic research, scholars have deeply analyzed the differences between developed and developing countries in their willingness to make voluntary carbon information disclosure. It is found that although developed countries have started earlier and are more perfect in carbon information disclosure system, developing countries are facing more urgent sustainable development challenges and transformation pressure, and their demand for carbon information is more urgent. This finding reveals the importance that developing countries place on high-quality carbon information in promoting a green and low-carbon transition. Further, when companies respond positively to this demand by proactively disclosing more and more detailed carbon information, their market value tends to increase significantly, which not only helps to enhance investor confidence, but also establishes a good image of the company itself as a responsible and trustworthy organization [6].

With the increasingly strict control of carbon emissions by the state, carbon disclosure has become the research focus of many scholars, and studies on the relationship between carbon disclosure and corporate financial performance have become increasingly abundant. Some scholars believe that enhanced carbon disclosure means that firms need to invest more resources in carbon emission reduction, which may increase the cost burden in the short term, thus negatively affecting financial performance [7,8]. On the other hand, another part of scholars believe that in the long run, good carbon disclosure can enhance corporate brand image [9], strengthen market competitiveness, attract more investors and consumers who are concerned about sustainable development, and then gain more advantages in the real economy, which will ultimately improve the financial performance of firms [10]. Therefore, the relationship between carbon disclosure and financial performance is inconclusive and deserves further exploration. Quantitatively examining the impact of carbon disclosure on corporate financial performance and an in-depth study of its underlying mechanism is of great significance in promoting the sound development of firms and realizing the goal of "double carbon". Based on the data of heavy pollution listed companies in Shanghai and Shenzhen A-shares from 2013 to 2023, this paper utilizes the signal theory, rent-seeking theory, and sustainable development theory to study the relationship between the carbon information disclosure and financial performance of firms, improve the credit ratings of firms in financial institutions such as banks, reduce the cost of debt financing of firms, and then enhance the financial performance of firms, and carbon trading policy positively moderates the impact of carbon disclosure on the financial performance of firms.

In exploring the relationship between carbon disclosure and corporate financial performance, there are several limitations in previous studies. First, most studies focus on analyzing the direct correlation between carbon disclosure and corporate financial performance, but rarely explore the complex mechanisms and transmission paths behind this association. This limitation of research orientation restricts our in-depth study of the relationship between the two. In addition, some of the studies are limited to the samples of enterprises within a specific geographical area and lack a broad perspective across geographical areas, which may weaken the general applicability and external validity of the research conclusions. Second, from the perspective of theory construction, the current theoretical framework on the relationship between carbon disclosure and corporate financial performance is still in the stage of construction and improvement, and has not yet formed a unified and solid theoretical foundation and explanatory mechanism. This status quo has led some studies to oversimplify the model setting, failing to comprehensively incorporate and consider a series of potential moderating

variables and intermediary factors, such as differences in enterprise size, industry-specific attributes, and changes in the external policy environment, which limits the model's explanatory power and predictive ability in complex realities. Finally, the difficulty of data acquisition is also one of the most important factors restricting the in-depth development of research in this area. Due to the fact that carbon disclosure data often involves non-financial information and different disclosure standards, as well as the high cost and time constraints of the data collection process, it is difficult for some studies to obtain sufficient and representative sample data. Such data limitations may not only introduce sample bias, but also lead to biased conclusions. This paper explores the micro-mechanisms of carbon disclosure on corporate financial performance by including institutional investors' shareholding ratio and debt financing cost in the scope of the study based on signaling theory, rent-seeking theory, and sustainable development theory. In order to alleviate the possible endogeneity problem and sample self-selection problem of this paper, this paper adopts the instrumental variable method and the method of excluding the sample of firms in the eastern region, respectively, to ensure the robustness of the conclusions of this paper. In addition, this paper examines the impact of carbon disclosure on financial performance in different regions based on China's national conditions and its unbalanced regional development, with a view to further refining the current study.The marginal contributions of this paper are: first, carbon disclosure has become an important research topic in the field of environmental accounting, and a high degree of disclosure may bring reputation enhancement to firms and strengthen their financial performance [7]. At the same time, it may increase the production cost of the firm and reduce the output efficiency of the firm, which in turn reduces the financial performance [6,11,12]. Although more scholars have studied the relationship between carbon disclosure and financial performance, they have reached different conclusions. Based on signaling theory, rent-seeking theory, and sustainable development theory, this paper explores the role of creditors, institutional investors, and other stakeholders in the relationship between carbon disclosure and corporate financial performance, which enriches the research perspectives and expands the existing research on the relationship between carbon disclosure and financial performance. Second, a heterogeneity test is conducted based on the regions of the sample firms to further analyze whether there are geographical differences in the relationship between carbon disclosure and corporate financial performance, which provides practical guidance for China to implement carbon disclosure policies by region, learn from other countries with advanced disclosure experiences, and promote the carbon trading market in the future to achieve the goal of carbon emission reduction.

## 2. Literature review

The accelerated pace of economic development has also generated more environmental pollution problems. In recent years, with the promulgation of a series of carbon-related policies in China, the research horizons of carbon disclosure in the academic community have become increasingly broad and the angles of discussion have become more and more diversified. Scholars have also conducted a lot of research on the influencing factors of carbon disclosure. From the macro level, government and investor pressure [13], the degree of legal system constraints [14], national culture [15] and so on will affect the quality of corporate carbon information disclosure. From the micro level, the political relevance of firms [16], the gearing ratio [17], the proportion of women in management [18], etc. will also affect the level of carbon information disclosure of firms to some extent. Especially with the deepening of national attention to environmental information disclosure in recent years, carbon disclosure and carbon management have gradually been linked to corporate strategy. However, in terms of the economic consequences of carbon disclosure, the academic community has not reached a

unified understanding of the relationship between carbon disclosure and financial performance, but most scholars believe that there is a positive correlation between the two.

Wang et al. found that firms increase the degree of environmental information disclosure, can broaden the analytical perspective of analysts, and then have a positive impact on financial performance [19]. Downar with the sample of listed companies in the United Kingdom, the study shows that carbon disclosure does not significantly weaken the financial performance of the firm, but it may still make the firm produce a certain operational pressure [20]. Bedi et al., based on an exhaustive data analysis of 100 firms in India over the period of 2018 to 2021, found that firms' implementation of carbon disclosure strategies can accelerate the process of global climate disclosure practices, and pointed out that there is a significant positive correlation between this initiative and the improvement of firms' financial performance. The study provides empirical evidence on how firms can enhance their competitiveness in the marketplace by being transparent about their environmental performance [21]. Focusing on Korean firms, Lee et al. found that voluntary carbon disclosure by firms can effectively reduce the risk of stock price crashes and help firms achieve more stable returns in the capital market. This finding demonstrates the positive role of carbon disclosure in maintaining investor confidence and stabilizing market sentiment [22]. In addition, Sun et al. further point out that improving the quality of carbon disclosure is a key way to alleviate the information asymmetry between firms and stakeholders. Through high-quality disclosure, companies can establish a more transparent communication mechanism and enhance stakeholders' awareness of and confidence in their environmental performance and long-term development prospects, thus indirectly contributing to the enhancement of corporate financial performance [23]. Nazila and Fauziah, based on a sample of 27 companies in Indonesia over a period of four years, demonstrate the positive impact of the comprehensiveness of carbon and social responsibility disclosure on corporate financial performance. They found that those firms that have a high level of carbon emission and social responsibility disclosure have a positive impact on corporate financial performance. They find that firms that exhibit high levels of both carbon and social responsibility tend to have superior financial performance. This finding emphasizes the importance of multidimensional disclosure in creating a positive corporate image and enhancing market recognition [24]. Chen et al. ESG disclosure can positively affect firm performance, especially for firms with strong reputational capital and high media attention, this positive effect is more significant [25]. Further, Hahn and Uyar et al. show that firms that make carbon disclosures provide advice to key stakeholders such as shareholders and creditors to make decisions that are favorable to the firms and strengthen the supervision of environmental protection of the firms by regulators, the public, and others to optimize the environmental performance of the firms [26,27]. Li and Liu et al. find that firms' carbon disclosure quality can reduce the cost of capital, and this negative correlation is more restrictive in firms with poorer carbon performance [11]. Liesen et al. points out that specifying disclosure standards through mandatory disclosure of carbon information will enhance market efficiency and help firms allocate more optimally in the real economy [10]. In addition, some scholars consider that carbon disclosure may have long-term effects and propose to examine the effectiveness of carbon disclosure from two perspectives, i.e., examining the current cash flow status of firms in the short term, and making a good valuation of future profitability in the long term [28].

However, some scholars believe that with the deepening of carbon disclosure, the financial performance of firms may suffer negative impacts and deterioration. The legitimacy theory requires firms to satisfy the implicit social contract of stakeholders to maintain their survival and development [29]. Therefore, when firms face more serious pollution problems, to maintain their legitimacy status, they will be forced to conduct information disclosure [30] to divert investors' attention from their behavior [31], and the effectiveness of information disclosure is

difficult to guarantee. When firms face government pressure to make carbon disclosure, firms with poor environmental performance may make involuntary and low-credibility disclosures, or may even make false disclosures, to salvage their image. Empirical studies have shown that firms with a high frequency of voluntary disclosure may have surplus manipulation before equity refinancing, and share prices and earnings will drop significantly after financing [32], which provides evidence of the economic consequences of voluntary disclosure by firms. In addition, the costs invested by firms to take on environmental responsibility will lead to the diversion of resources, affecting the improvement of firms' financial performance.

Siddique et al. using a sample of global firms as a research sample, found that, in the short run, enhanced carbon disclosure by firms may result in a negative trend due to the temporary decline of financial performance as a result of the initial investment costs, management adjustments, and other factors. However, in the long run, carbon disclosure can enhance corporate image, increase investor confidence, promote effective allocation of resources, and ultimately have a positive impact on financial performance [9]. Plumlee et al. found through an empirical study that, in the short run, carbon disclosure will be accompanied by additional cost inputs, regulatory pressures, and uncertainty of market responses, and this behavior will instead lead to a corporate financial performance decline [8].

By analyzing the existing literature, it can be found that with the rapid development of the economy and the increasing prominence of environmental pollution, carbon disclosure has become the focus of academic attention. The research perspectives are gradually diversified, covering multiple levels such as government policy, legal system, corporate culture, and internal corporate governance structure. Although most scholars agree that carbon disclosure is positively related to financial performance, and emphasize its positive effects on corporate image, market trust, and resource allocation, there are still differences in academic perceptions of this relationship. Some studies point out that in the short term, carbon disclosure may put pressure on corporate finances due to increased costs and market uncertainty, i.e., there is a negative correlation between carbon disclosure and financial performance.

## 3. Theory and hypothesis

Scholars who hold the signaling theory believe that carbon disclosure, as an important means of communicating non-financial information, can effectively enhance enterprise value [33]. Specifically, a high-quality and transparent carbon disclosure system can reduce the information asymmetry that exists between enterprises and stakeholders, and enhance stakeholders' trust in the enterprise, which is not only the basis for economic transactions, but also the key to the realization of sustainable development of enterprises.Further, the theory of sustainable development, as an important theoretical basis for modern enterprise management and strategic planning, emphasizes that enterprises must take into account the protection of the ecological environment and the fulfillment of social responsibility while pursuing economic growth, and abandon the traditional narrow concept of environmental protection and economic development as a zero-sum game. Under this theoretical framework, through the implementation of carbon disclosure strategy, enterprises actively release to the market their commitment to environmental protection and actively practice social responsibility signals. This not only demonstrates the company's deep recognition of the green, low-carbon and recycling development model, but also helps to build a positive green brand image and enhance market attractiveness, which in turn attracts investors and partners who also uphold the concept of sustainable development. As an informal system, the deepening of social trust reduces the friction in the transaction process between the two parties [34], which enables firms to reach cooperation at a lower cost, which in turn helps them to improve their internal and external financing capacity

[35], and ultimately grows the financial performance of the firms that make carbon disclosures. On the contrary, if firms do not make disclosures, the degree of information asymmetry between firms and stakeholders is exacerbated [36]. As it is difficult for investors to obtain data on the environmental activities carried out by firms, it is more difficult to assess the risks and benefits arising from the firms' activities. As a result, investors need to spend a large cost to search for information on the environmental activities of firms, and this cost will eventually be borne by firms that do not disclose information [37], which ultimately affects the financial performance of firms.

Also supporting this view is the rent-seeking theory, which explains the positive relationship between the quality of carbon disclosure and financial performance from the perspective of social responsibility. Rent-seeking activity refers to the process in which firms utilize their association with the government to obtain excess "rent" income in the process of government control and intervention in market economic activities [38]. Through active rent-seeking activities, firms' legitimacy is enhanced [39], which effectively establishes a strong government-enterprise relationship and leads to more financial subsidies and key resource support from the government [40]. In addition, these activities can, to a certain extent, mitigate the negative impact of market failure on business operations and ensure that the stability of corporate finance is effectively maintained [41]. It has been found that corporate social responsibility is a kind of rent-seeking behavior [42]. By fulfilling social responsibility, firms form a good reputation in front of the public, which can not only help firms reduce the cost of debt financing [43,44], and show the government the contribution made by the enterprise to society, and form an implicit exchange of benefits with the government to obtain more resources to support the enterprise [39]. Comparatively, the resources allocated by local governments will be tilted to firms with a higher degree of social responsibility fulfillment [42], and the external disclosure of environmental information is precisely the social responsibility that firms must fulfill. Therefore, carbon disclosure by firms can be regarded as firms seeking rent from the government to seek some benefits on the economic level, such as giving firms appropriate tax breaks and corresponding financial subsidies [42], and the tilting of these financial resources enhances the financial performance of firms.

In summary, from the perspective of signaling theory, sustainable development theory, and rent-seeking theory, the disclosure of carbon information by enterprises can reduce the degree of information asymmetry between enterprises and stakeholders, enhance the trust of outsiders in the enterprise, obtain better quality resources, improve financial performance and realize sustainable development; enterprises, for rent-seeking, take on a certain degree of environmental protection responsibility for the local government and can obtain a tilt of the policy. Thereby enhancing the financial performance of enterprises. Based on this, this paper puts forward the following hypotheses:

H1: Other things being equal, there is a positive contribution of firms' carbon disclosure to firms' financial performance.

## 4. Methodology

### 4.1 Sample selection and data source

This paper selects the heavy pollution enterprises listed in Shanghai and Shenzhen A-shares from 2013 to 2023. The data in this paper are selected from 2013 because since 2012, after the "The Five-sphere Integrated Plan" was put forward in the report of the "18th National Congress", enterprises gradually began to pay attention to the disclosure of environmental information, and at this time, the environmental protection data are easier to obtain and have

reference significance. A total of 18 industries, including coal mining and washing, oil and gas mining, ferrous metal mining and processing, non-ferrous metal mining and processing, textile, leather, fur, feather and their products and footwear, are defined in this article as heavy polluting industries. According to this criterion, a total of 7965 observations of 1065 listed companies are screened out in this paper, and 2084observations of 339 listed companies remain after excluding samples with missing certain indicators, excluding ST samples, financial enterprise samples, and samples with missing data, and shrinking the upper and lower 1% tail of the main variables. The corporate financial data are from the Cathay Pacific (CSMAR) database.

## 4.2 Measurement model

To test hypotheses1, the model is designed as follows:

$$Roa_{i,t} = \beta_0 + \beta_1 Cdi_{i,t} + \beta_2 Tang_{i,t} + \beta_3 Soe_{i,t} + \beta_4 Size_{i,t} + \beta_5 Growth_{i,t} +$$
$$\beta_6 Intang_{i,t} + \beta_7 Invent_{i,t} + \beta_8 Ltd_{i,t} + \Sigma Industry + \Sigma Year + \varepsilon_{i,t}$$

(1)

Where $Roa_{i,t}$ is the financial performance of firm i in year t, and $Cdi_{i,t}$ is the carbon disclosure of firm i in year t. In the model, $\beta_0$ is the constant term, $\beta_i$ is the regression coefficients of each variable, and $\varepsilon$ is the random perturbation term and controls for year and industry fixed effects. If the empirical results show $\beta_1$ is positive, then H1 is proved, that is, carbon disclosure positively affects corporate financial performance.

## 4.3 Variable definition

**4.3.1 Dependent variable.** In this paper, Roa (net profit margin of total assets) is selected as the core proxy variable. Roa is chosen as a proxy variable for the financial performance of the enterprise, which considers the efficiency of the assets and can measure the operating results of the enterprise in the current period due to carbon disclosure, and can reflect the profitability and production capacity of the enterprise in the current period. In addition, compared with other financial indicators, Roa is a short-term value indicator, and its results can more accurately measure the short-term financial performance improvement brought by enterprises to improve the quality of carbon disclosure. Therefore, this paper chooses Roa as a proxy variable for financial performance.

**4.3.2 Independent variable.** Referring to Li Li's (2019) study [32], this paper constructs a system for evaluating corporate carbon disclosure. Using Cdi as a proxy variable for measuring corporate carbon disclosure, a total of four aspects are evaluated carbon emission reduction strategy and management, carbon emission reduction accounting, carbon emission reduction measures, and carbon emission reduction performance, and the specific scoring items are shown in Table 1. Final application of the entropy method to construct a composite indicator for carbon disclosure.

**4.3.3 Control variables.** We control the correlation variables concerning the studies of Wen Subin [45], and Song Xiaohua [46]. The relevant variables are shown in Table 2.

## 5. Empirical analysis

## 5.1 Descriptive statistics

According to Table 3, the maximum value of financial performance (Roa) is 0.204, the minimum value is -0.204, the standard deviation is 0.055, and the mean value is 0.036, which indicates that there is a large difference in profitability among Chinese listed heavy polluters and that the overall level of input-output in this industry is low. The mean and median of carbon

Table 1. Carbon disclosure indicators.

| Level 1 indicators | Secondary indicators | explain |
|---|---|---|
| Carbon emission reduction strategy and management | Carbon emission reduction institutions setting up | An organization is set up to get 1 point, otherwise 0 points |
| | Strategy and goal | 1 for qualitative description, 2 for quantitative description, otherwise 0 |
| | Investment in energy conservation projects | |
| | The implementation of the national emission reduction policies | |
| | Enterprise internal environmental protection assessment system | |
| | Solid waste discharge volume | |
| Carbon emission reduction accounting | Accounting for greenhouse gas emissions | |
| | Accounting for greenhouse gas emissions | |
| | Accounting for the emission intensity of major pollutants | |
| | Report the main distribution of emissions | |
| Carbon emission reduction measures | Purchase and upgrade environmental protection facilities | |
| | Develop low-carbon technology and low-carbon products | |
| | Research and development of pollution prevention technology and process | |
| | Pollution monitoring equipment | |
| Carbon emission reduction performance | Whether the subject responsibility is clear | |
| | Emission-reduction performance appraisal method | |
| | Financial subsidies for carbon emission reduction | 1 point for the subsidy that year, otherwise 0 points |

disclosure (Cdi) are 0.090 and 0.063, with the minimum and maximum values of 0.003 and 0.400, respectively, indicating that the quality of disclosure is mixed and varies greatly among different enterprises. In addition, the rest of the variables are in the reasonable range.

## 5.2 Correlation analysis

By calculating the Pearson correlation coefficient, Table 4 can be obtained. According to the Table 4, the correlation coefficient between Roa and Cdi is 0.086, which is significant at the 1% level, which indicates that corporate carbon disclosure has a significant positive impact on financial performance in the whole sample, and this finding is the same as the existing literature [19,21–25]. In addition, the absolute value of the correlation coefficients of the variables is

Table 2. Description of variables.

| Type of variable | Variable name | variable symbol | Variable measure |
|---|---|---|---|
| Independent variable | Financial performance | Roa | Ratio of net profit after tax to average total assets |
| Dependent variable | Carbon Disclosure | Cdi | The methodology for constructing the indicators is shown above |
| Control variable | Asset structure | Tang | (net fixed assets + net inventories)/total assets |
| | Nature of property rights | Soe | State-owned enterprises are assigned a value of 1, otherwise 0 |
| | Enterprise size | Size | Logarithm of total assets at the end of the period |
| | Enterprise growth capacity | Growth | Revenue growth rate |
| | Intangible asset intensity | Intang | Net intangible assets to total assets at the end of the period |
| | Inventory intensity | Invent | Inventory balance to total assets at the end of the period |
| | Long-term debt ratio | Ltd | Long-term liabilities to total assets |
| | Year | Year | Dummy variables representing years |
| | Sector | Industry | Dummy variables representing industries |

**Table 3. Descriptive statistics.**

| Variable | N | Mean | SD | Min | p25 | p50 | p75 | Max |
|---|---|---|---|---|---|---|---|---|
| Roa | 2084 | 0.036 | 0.055 | -0.204 | 0.011 | 0.032 | 0.063 | 0.204 |
| Cdi | 2084 | 0.090 | 0.082 | 0.003 | 0.029 | 0.063 | 0.126 | 0.400 |
| Tang | 2084 | 0.470 | 0.162 | 0.068 | 0.354 | 0.475 | 0.589 | 0.796 |
| Size | 2084 | 22.930 | 1.514 | 18.910 | 21.710 | 22.890 | 24.010 | 25.920 |
| Growth | 2084 | 0.092 | 0.405 | -0.684 | -0.081 | 0.035 | 0.179 | 3.816 |
| Intang | 2084 | 0.055 | 0.052 | 0.001 | 0.024 | 0.040 | 0.066 | 0.339 |
| Invent | 2084 | 0.103 | 0.088 | 0.001 | 0.038 | 0.085 | 0.141 | 0.408 |
| Ltd | 2084 | 0.138 | 0.122 | 0.000 | 0.041 | 0.109 | 0.198 | 0.505 |

less than 0.5, which indicates that there is no serious multicollinearity relationship among the variables. For the variables whose absolute values of the correlation coefficients in Table 4 are between 0.3 and 0.6, this paper further conducts the VIF test. The results show that the VIF values of all independent variables are less than 5, which means that there is no multicollinearity problem.

## 5.3 Main effect regression

To test the relationship between carbon disclosure quality and corporate financial performance, this paper takes the Roa of the enterprise in the current period as the explanatory variable and the quality of corporate carbon disclosure as the explanatory variable, and the results are shown in Table 5. In Table 5, industry and year-fixed effects are controlled respectively, and the results are all significant, this regression result verifies H1, i.e., the enhancement of the quality of carbon disclosure can improve corporate financial performance. Taking column (3) as an example, when Cdi increases by one standard deviation, Roa is enhanced by about 11.39% (0.082*0.050/0.036), which is significant in terms of economic effect, and H1 is verified.

## 5.4 Robustness test

**5.4.1 Instrumental variable approach.** Since there may be endogenous causality between carbon disclosure quality and firms' financial performance, i.e., the growth of firms' financial performance may motivate firms to improve their carbon disclosure quality. To alleviate the problems of endogeneity, omitted variables, and sample self-selection, this paper uses the instrumental variable method for regression analysis. For the selection of instrumental

**Table 4. Correlation analysis.**

| | Roa | Cdi | Tang | Soe | Size | Growth | Intang | Invent | Ltd | VIF |
|---|---|---|---|---|---|---|---|---|---|---|
| Roa | 1 | | | | | | | | | |
| Cdi | 0.086*** | 1 | | | | | | | | 1.19 |
| Tang | -0.191*** | 0.065*** | 1 | | | | | | | 1.55 |
| Soe | -0.214*** | 0.079*** | 0.298*** | 1 | | | | | | 1.48 |
| Size | 0.064*** | 0.362*** | 0.271*** | 0.315*** | 1 | | | | | 1.64 |
| Growth | 0.056** | -0.038* | -0.130*** | 0.004 | -0.145*** | 1 | | | | 1.07 |
| Intang | 0.002 | 0.070*** | -0.240*** | 0.062*** | 0.079*** | 0.035 | 1 | | | 1.55 |
| Invent | -0.089*** | -0.010 | 0.179*** | -0.114*** | -0.010 | 0.027 | -0.143*** | 1 | | 1.56 |

Notes: *p < 0.1; **p < 0.05; ***p < 0.01.

**Table 5. Carbon disclosure and corporate financial performance.**

|  | Roa | | |
| --- | --- | --- | --- |
|  | (1) | (2) | (3) |
| Cdi | 0.050*** | 0.049*** | 0.050*** |
|  | (0.013) | (0.013) | (0.012) |
| Tang | -0.029*** | -0.040*** | -0.035*** |
|  | (0.009) | (0.009) | (0.009) |
| Soe | -0.023*** | -0.025*** | -0.024*** |
|  | (0.003) | (0.003) | (0.003) |
| Size | 0.008*** | 0.010*** | 0.009*** |
|  | (0.001) | (0.001) | (0.001) |
| Growth | 0.010*** | 0.011*** | 0.011*** |
|  | (0.003) | (0.003) | (0.003) |
| Intang | -0.069*** | -0.006 | 0.001 |
|  | (0.022) | (0.025) | (0.025) |
| Invent | -0.123*** | -0.094*** | -0.097*** |
|  | (0.014) | (0.016) | (0.015) |
| Ltd | -0.121*** | -0.161*** | -0.157*** |
|  | (0.011) | (0.013) | (0.013) |
| Industry fixed effects | NO | YES | YES |
| Year fixed effects | NO | NO | YES |
| Constant | -0.087*** | -0.138*** | -0.123*** |
|  | (0.020) | (0.023) | (0.023) |
| Observations | 2084 | 2084 | 2084 |
| Adjusted $R^2$ | 0.156 | 0.178 | 0.193 |

Notes

*$p < 0.1$

**$p < 0.05$

***$p < 0.01$(robust standard errors adjusted for heteroscedasticity are reported in parentheses).

variables, this paper takes the average value of carbon disclosure in the year in the province to which the sample enterprise belongs (ACdi) as instrumental variables. If the quality of carbon disclosure in the year in the province to which the enterprise belongs is high, it will drive the enterprise to improve the quality of carbon disclosure in the year, with a smaller impact on the enterprise's financial performance. Therefore, the selection of instrumental variables meets the requirements. The regression results are shown in Table 6. Columns (1) and (2) show the regression results of taking the average value of carbon disclosure in the year (ACdi) as an instrumental variable in the provinces where the sample firms belong. The first-stage regression results show that ACdi is positively correlated with the explanatory variable Cdi at the 1% level, as expected. Column (2) shows the results of the second-stage regression, which is significantly positive at the 10% level, and the Cragg-Donald Wald F-value is 289.01, rejecting the hypothesis of "weak instrumental variable". The Kleibergen-Paap rk LM value is 93.56 which is significant at the 1% level. Therefore, after correcting for the endogeneity problem, the conclusion of this paper is still valid.

**5.4.2 Explanatory variables lagged one period.** Some studies suggest that larger or better-performing firms have stronger incentives to make disclosures [47–49], i.e., there may be a reverse causation problem between the two. To solve the possible reverse causality problem in the subtext, this paper uses the carbon disclosure lagged by one periods instead of the current

**Table 6. Instrumental variables regression results.**

| | (1) | (2) |
|---|---|---|
| | **Cdi** | **Roa** |
| | **Phase I** | **Phase II** |
| **ACdi** | 0.837*** | |
| | (16.35) | |
| **Cdi** | | 0.136*** |
| | | (3.01) |
| **Tang** | 0.007 | -0.034*** |
| | (0.58) | (-3.77) |
| **Soe** | -0.008** | -0.024*** |
| | (-2.06) | (-8.13) |
| **Size** | 0.015*** | 0.008*** |
| | (11.22) | (5.94) |
| **Growth** | 0.003 | 0.010*** |
| | (0.96) | (3.37) |
| **Intang** | 0.026 | -0.003 |
| | (0.65) | (-0.10) |
| **Invent** | 0.036* | -0.102*** |
| | (1.65) | (-6.44) |
| **Ltd** | 0.012 | -0.160*** |
| | (0.65) | (-12.39) |
| **Industry fixed effects** | YES | YES |
| **Year fixed effects** | YES | YES |
| **Constant** | -0.330*** | -0.072*** |
| | (-11.35) | (-2.86) |
| **Observations** | 2,084 | 2,084 |
| **R²** | | 0.188 |
| **Phase I F-value** | 267.19 | |
| **Kleibergen-Paap rk LM value** | 93.56*** | |
| **Cragg-Donald Wald F value** | 289.01 | |

Notes

*$p < 0.1$

**$p < 0.05$

***$p < 0.01$ (robust standard errors adjusted for heteroscedasticity are reported in parentheses).

period to conduct the regression, and the results, as shown in column (1) of Table 7, are still significantly positively correlated, so the conclusion of this paper is robust.

**5.4.3 Shorter sample periods.** To eliminate the impact of the new crown epidemic on firms' operations, this paper excludes data from 2020–2023 to ensure the robustness of the regression results. The results, as shown in column (2) of Table 7, show that the relationship between carbon disclosure quality and firms' financial performance remains significantly positive. Therefore, excluding the effect of the new crown epidemic, the improvement of carbon disclosure quality can still promote the growth of corporate financial performance.

**5.4.4 Substitution of explanatory variables.** To ensure that the results are robust, this paper uses Roe as a proxy variable for Roa in the regression, and the regression results, as shown in column (3) of Table 7, show that Cdi is still positively correlated with Roe, indicating that the results are robust.

**Table 7. Robustness test results.**

| | (1) | (2) | (3) | (4) | | (5) |
|---|---|---|---|---|---|---|
| | **Roa** | **Roa** | **Roe** | **Roa** | | **Roa** |
| | | | | carbon pilot | Non-carbon pilot | |
| **L2Cdi** | 0.048*** | | | | | |
| | (0.015) | | | | | |
| **Cdi** | | 0.030* | 0.110*** | 0.064*** | 0.049*** | 0.107*** |
| | | (0.017) | (0.031) | (0.021) | (0.016) | (0.021) |
| **Tang** | -0.029*** | -0.037*** | -0.096*** | -0.016 | -0.056*** | -0.062*** |
| | (0.010) | (0.011) | (0.028) | (0.015) | (0.012) | (0.016) |
| **Soe** | -0.025*** | -0.030*** | -0.061*** | -0.027*** | -0.023*** | -0.030*** |
| | (0.003) | (0.004) | (0.008) | (0.005) | (0.004) | (0.006) |
| **Size** | 0.010*** | 0.009*** | 0.025*** | 0.008*** | 0.010*** | 0.011*** |
| | (0.001) | (0.001) | (0.003) | (0.002) | (0.001) | (0.002) |
| **Growth** | 0.013*** | 0.009** | 0.021** | 0.007 | 0.011*** | 0.011** |
| | (0.004) | (0.003) | (0.009) | (0.006) | (0.004) | (0.005) |
| **Intang** | 0.004 | -0.020 | -0.019 | 0.035 | -0.024 | -0.038 |
| | (0.027) | (0.029) | (0.064) | (0.041) | (0.033) | (0.034) |
| **Invent** | -0.092*** | -0.084*** | -0.109*** | -0.113*** | -0.077*** | -0.084*** |
| | (0.017) | (0.019) | (0.042) | (0.025) | (0.020) | (0.028) |
| **Ltd** | -0.162*** | -0.157*** | -0.284*** | -0.168*** | -0.142*** | -0.148*** |
| | (0.014) | (0.016) | (0.042) | (0.020) | (0.016) | (0.019) |
| **Industry fixed effects** | YES | YES | YES | YES | YES | YES |
| **Year fixed effects** | YES | YES | YES | YES | YES | YES |
| **Constant** | -0.145*** | -0.124*** | -0.399*** | -0.104*** | -0.131*** | -0.142*** |
| | (0.026) | (0.030) | (0.060) | (0.037) | (0.032) | (0.038) |
| **Observations** | 1649 | 1230 | 2078 | 767 | 1317 | 812 |
| **Adjusted R²** | 0.181 | 0.223 | 0.126 | 0.172 | 0.229 | 0.291 |

Notes

*p < 0.1

**p < 0.05

***p < 0.01 (robust standard errors adjusted for heteroscedasticity are reported in parentheses).

**5.4.5 Excluding the impact of carbon trading policies.** China started to pilot carbon emissions trading policies in 2013, and pilot cities generally adopt more stringent regulatory mechanisms and information disclosure policies. Considering the impact of this policy on the regression results, this paper conducts regressions on the subsamples of firms in carbon trading pilot cities and firms not in carbon trading pilot cities separately. The results are shown in column (4) of Table 7, the regression coefficients of carbon disclosure are all significantly positive, and there is no significant difference between the results in the two subsamples. It can be seen that the conclusions remain robust after considering the exogenous shock of the carbon trading policy.

**5.4.6 Exclusion of enterprises located in the Eastern Region.** The carbon disclosure behavior of enterprises is geographically inclined. Since the eastern region is more economically developed and has a relatively better legal environment, the intensity of environmental regulation in this region is higher, and the requirements for carbon disclosure are also relatively higher. To alleviate the problem caused by "sample self-selection", this paper conducts a regression after excluding the samples of firms located in the eastern region, and the results, as

shown in column (5) of Table 7, show that there is still a positive correlation between carbon disclosure and firms' financial performance, which suggests that this paper does not suffer from serious self-selection problems.

### 5.5 Mechanism analysis

**5.5.1 Intermediation of institutional investors' shareholdings.**   Under the trend that all parties in society pay more and more attention to environmental protection, the capital market also pays more attention to the fulfillment of corporate social responsibility, and investors will have a greater possibility to invest in firms with a high degree of social responsibility fulfillment. Carbon disclosure, as an important part of corporate social responsibility fulfillment, is also a concern for various types of investors [50]. Compared with individual investors, institutional investors have more specialized knowledge reserves, richer information sources, and a more comprehensive interpretation of the national green economy development trend, and therefore are more likely to notice the firms' investment in social responsibility and information disclosure [51] and thus make investment decisions.

From the perspective of principal-agent theory, first of all, institutional investors, as external stakeholders of firms, are affected by the quality of corporate disclosure in their investment decisions. In the principal-agent relationship, due to information asymmetry and conflict of interest, the agent may take behaviors that are inconsistent with the interests of the principal, which increases the agency cost. When companies disclose high-quality carbon information, institutional investors, as external stakeholders of companies, can better understand the environmental management status and potential risks of companies, which reduces the degree of information asymmetry. This not only helps institutional investors to more accurately assess the value and risk of the company and reduce the agency costs arising from insufficient information but also increases their investment confidence in the firm, which in turn leads to more informed investment decisions and higher shareholding ratios. Second, institutional investors, as external shareholders, have a stronger incentive and ability to obtain internal information about the firm, monitor the decisions and behaviors of corporate executives, inhibit inefficient investment [52], and prompt the firm to act by the principle of maximizing shareholders' interests. This helps to reduce moral hazard and conflict of interest within the firm, improve the efficiency of corporate governance, and then increase the financial performance of the firm. Finally, improving the quality of carbon information disclosure can enhance the reputation of the firm, which helps attract more institutional investors to pay attention to and invest in the firm. Therefore, improving the quality of carbon disclosure is not only conducive to the transparency of its environmental management but also an important means of attracting institutional investors and optimizing the investment structure.

**5.5.2 Intermediation of debt financing costs.**   Principal-agent theory points out that, in the environment of a market economy, there exists the phenomenon of information asymmetry between the firm and the stakeholders, due to the separation of ownership and operation rights, the firm operator tends to grasp more internal information, making it difficult for stakeholders to have a comprehensive insight into the business reality of the firm, which is easy to induce the risk of adverse selection. At the same time, as an agent of the firm operators, in the process of pursuing their interests, they may ignore or harm the rights and interests of stakeholders, which in turn affects the effective allocation of market resources and increases the cost of firm financing. However, the improvement of the quality of corporate disclosure can reduce the degree of information asymmetry between firms and stakeholders [36], help stakeholders understand the risks and performance of firms, and thus make it easier for firms to obtain financing [35].

First, from the principal-agent theory, agency costs include monitoring costs, incentive costs, and residual losses. When firms improve the quality of carbon disclosure, it becomes easier for creditors to supervise the environmental management of firms, thus reducing the supervision cost. At the same time, creditors can also more accurately assess the environmental risks and sustainability of the firm, thus designing incentive contracts more rationally and reducing incentive costs. All these help to reduce the agency cost of firms and thus the cost of debt financing. Second, the quality of corporate disclosure is likewise an important aspect of its fulfillment of social responsibility. By fulfilling social responsibility, the reputation and corporate value of firms are enhanced, stabilizing the psychological expectations of both investment and financing parties, reducing friction in the transaction process, and lowering the costs required to reach cooperation [53]. For example, banks' trust in firms can help firms obtain lower loan interest rates [54]. As a result, firms that have a high reputation through disclosure have relatively low debt financing costs. Finally, corporate disclosure of carbon information is not only a concrete manifestation of its social responsibility but also a strategy to seek resource support from the government. Through the disclosure of social responsibility information, firms can establish closer ties with the government and all walks of life, and obtain more policy and resource support, which can help them break through development bottlenecks and obtain more convenient financing opportunities. At the same time, the social responsibility performance of firms also affects the audit results of the regulatory authorities, which in turn affects their financing costs [54].

Based on the above analysis, firms that improve the quality of carbon disclosure have the potential to both reduce the cost of debt financing they face and increase their institutional investor shareholding. To verify this relationship, this paper designs the following model:

$$Roa_{i,t} = \beta_0 + \beta_1 Cdi_{i,t} + \beta_2 Tang_{i,t} + \beta_3 Soe_{i,t} + \beta_4 Size_{i,t} + \beta_5 Growth_{i,t} +$$
$$\beta_6 Intang_{i,t} + \beta_7 Invent_{i,t} + \beta_8 Ltd_{i,t} + \Sigma Industry + \Sigma Year + \varepsilon_{i,t} \tag{2}$$

$$Mediate_{i,t} = \alpha_0 + \alpha_1 Cdi_{i,t} + \alpha_2 Tang_{i,t} + \alpha_3 Soe_{i,t} + \alpha_4 Size_{i,t} + \alpha_5 Growth_{i,t} +$$
$$\alpha_6 Intang_{i,t} + \alpha_7 Invent_{i,t} + \alpha_8 Ltd_{i,t} + \Sigma Industry + \Sigma Year + \varepsilon_{i,t} \tag{3}$$

$$Roa_{i,t} = \gamma_0 + \gamma_1 Cdi_{i,t} + \gamma_2 Mediate_{i,t} + \gamma_3 Tang_{i,t} + \gamma_4 Soe_{i,t} + \gamma_5 Size_{i,t} +$$
$$\gamma_6 Growth_{i,t} + \gamma_7 Intang_{i,t} + \gamma_8 Invent_{i,t} + \gamma_9 Ltd_{i,t} + \Sigma Industry + \Sigma Year + \varepsilon_{i,t} \tag{4}$$

Among them, model (2) is the same as the model (1), and $Mediate_{i,t}$ in the model (3) is the mediating variable. To verify the mechanism of carbon disclosure quality on corporate financial performance, this paper selects two indicators of corporate debt financing cost (Liability) and institutional investor shareholding ratio (INVH) as mediating variables. Among them, Liability refers to the practice of Yiling Wang, which is measured by the ratio of the sum of interest expense, fees, and other finance charges in finance costs to total liabilities of the enterprise at the end of the period [55]; $INVH_{i,t}$ is the proportion of institutional investors' shareholding of firm i in year t. The regression results are shown in Table 8. The regression results are shown in Table 8. In the analysis of the mediating role of debt financing cost, Liability is significantly negatively related to Cdi; after Liability is added to the regression, Cdi is significantly positively related to Roa, and Liability is significantly negatively related to Roa. In the analysis of the mediating role of institutional investors' shareholding, INVH is significantly positively related to Cdi; after INVH is added to the regression, Cdi is significantly positively related to Roa, and INVH is significantly negatively related to Roa. This indicates that the cost

**Table 8. Channel tests.**

| | Institutional investor holdings | | | Cost of debt financing | | |
|---|---|---|---|---|---|---|
| | Roa | INVH | Roa | Roa | Liability | Roa |
| Cdi | 0.050*** | 0.137*** | 0.048*** | 0.050*** | -0.028*** | 0.046*** |
| | (0.012) | (0.047) | (0.013) | (0.012) | (0.007) | (0.013) |
| INVH | | | 0.016** | | | |
| | | | (0.006) | | | |
| Liability | | | | | | -0.105* |
| | | | | | | (0.058) |
| Tang | -0.035*** | 0.037 | -0.035*** | -0.035*** | -0.001 | -0.033*** |
| | (0.009) | (0.028) | (0.009) | (0.009) | (0.006) | (0.009) |
| Soe | -0.024*** | 0.131*** | -0.026*** | -0.024*** | 0.002 | -0.024*** |
| | (0.003) | (0.010) | (0.003) | (0.003) | (0.002) | (0.003) |
| Size | 0.009*** | 0.059*** | 0.008*** | 0.009*** | 0.001 | 0.009*** |
| | (0.001) | (0.003) | (0.001) | (0.001) | (0.001) | (0.001) |
| Growth | 0.011*** | 0.017* | 0.010*** | 0.011*** | -0.001 | 0.010*** |
| | (0.003) | (0.010) | (0.003) | (0.003) | (0.003) | (0.003) |
| Intang | 0.001 | -0.140 | 0.003 | 0.001 | -0.025* | 0.003 |
| | (0.025) | (0.088) | (0.025) | (0.025) | (0.014) | (0.025) |
| Invent | -0.097*** | -0.300*** | -0.093*** | -0.097*** | -0.032*** | -0.094*** |
| | (0.015) | (0.053) | (0.015) | (0.015) | (0.009) | (0.015) |
| Ltd | -0.157*** | -0.039 | -0.156*** | -0.157*** | -0.037*** | -0.152*** |
| | (0.013) | (0.041) | (0.013) | (0.013) | (0.006) | (0.013) |
| Industry fixed effects | YES | YES | YES | YES | YES | YES |
| Year fixed effects | YES | YES | YES | YES | YES | YES |
| Constant | -0.123*** | -0.827*** | -0.110*** | -0.123*** | 0.056*** | -0.119*** |
| | (0.023) | (0.071) | (0.023) | (0.023) | (0.014) | (0.023) |
| Observations | 2084 | 2084 | 2084 | 2084 | 2021 | 2021 |
| Adjusted R² | 0.193 | 0.425 | 0.195 | 0.193 | 0.059 | 0.187 |

Notes

*p < 0.1

**p < 0.05

***p < 0.01 (robust standard errors adjusted for heteroscedasticity are reported in parentheses).

of debt financing and institutional investors' shareholding play a mediating role between carbon disclosure quality and corporate financial performance.

## 5.6 Further analysis

To further refine the study of the relationship between carbon disclosure and financial performance, this paper takes the region where the sample firms are registered as the criterion and divides all the firms into eastern, central, and western regions for group regression, and the regression results are shown in Table 9. The positive correlation between carbon disclosure and financial performance is significant in the samples of firms in the central region and the western region, while it is not significant in the samples of firms in the eastern region, and there may be several reasons for this difference. First, the economic development of the central and western regions is relatively lagging behind, and enterprises may still be in their infancy in terms of environmental awareness and carbon reduction technology. Therefore, when enterprises begin to pay attention to and disclose carbon information, this behavior may be

**Table 9. Heterogeneity test.**

| | Roa | | |
|---|---|---|---|
| | Eastern Region | Central Region | Western Region |
| Cdi | 0.027 | 0.072* | 0.116*** |
| | (0.018) | (0.038) | (0.041) |
| Tang | -0.028*** | -0.021 | -0.088*** |
| | (0.010) | (0.023) | (0.020) |
| Soe | -0.018*** | -0.029*** | -0.033*** |
| | (0.003) | (0.007) | (0.009) |
| Size | 0.009*** | 0.011*** | 0.012*** |
| | (0.001) | (0.002) | (0.003) |
| Growth | 0.008** | 0.018*** | 0.008 |
| | (0.003) | (0.006) | (0.007) |
| Intang | 0.018 | 0.006 | -0.021 |
| | (0.035) | (0.056) | (0.063) |
| Invent | -0.073*** | -0.092*** | -0.076 |
| | (0.020) | (0.034) | (0.047) |
| Ltd | -0.147*** | -0.204*** | -0.121*** |
| | (0.017) | (0.031) | (0.026) |
| Industry fixed effects | YES | YES | YES |
| Year fixed effects | YES | YES | YES |
| Constant | -0.124*** | -0.162*** | -0.183*** |
| | (0.027) | (0.050) | (0.059) |
| Observations | 1272 | 400 | 360 |
| Adjusted R² | 0.136 | 0.283 | 0.334 |

Notes

*p < 0.1

**p < 0.05

***p < 0.01 (robust standard errors adjusted for heteroscedasticity are reported in parentheses).

regarded as an important step for enterprises to actively fulfill their social responsibility and enhance their environmental image, which helps to enhance their sense of social responsibility and brand image, reduce information asymmetry [56], and thus make it easier to gain the recognition and support of investors, consumers, and other stakeholders, and to reduce the cost of equity capital for enterprises [57],which in turn positively affects financial performance. Second, the heavy pollution industries in the central and western regions are more developed, and the government will promote enterprises to strengthen carbon disclosure and carbon emission reduction through policy incentives and guidance. Such policy incentives may enable enterprises to obtain more tax incentives and government subsidies by disclosing carbon information, thus promoting the improvement of financial performance. Finally, in the eastern region, due to the more mature economic development, enterprises may face a more complex market environment and higher regulatory requirements, which makes enterprises in carbon disclosure has formed a more perfect system and norms, so the marginal impact of carbon disclosure on financial performance may be relatively small. Moreover, the market competition in the eastern region is more intense, and in order to maintain competitiveness in the market, enterprises may pay more attention to technological innovation, product upgrading and brand building, thus reducing the attention to carbon disclosure, resulting in the insignificant impact of carbon disclosure on financial performance.

## 6. Conclusions

To vigorously promote the realization of the "double carbon" goal, the State implements strict management of carbon emissions by enterprises, and at the same time introduces various policies to improve the level and quality of carbon information disclosure. However, is carbon disclosure contrary to the goal of corporate profitability, i.e., does corporate carbon disclosure behavior affect corporate financial performance? Based on this question, this paper focuses on the two dimensions of institutional investors' shareholding ratio and debt financing cost and analyzes how carbon disclosure affects firms' financial performance through these two critical paths in a China-specific context, using a sample of Chinese A-share listed firms in Shanghai and Shenzhen from 2013 to 2023 in the heavy polluting industry. It is found that the improvement of carbon disclosure quality enhances firms' financial performance, and this result is still significant after considering the endogeneity issue and conducting robustness tests. This paper further investigates the mechanism by which the quality of carbon disclosure positively affects corporate financial performance and finds that improving the quality of carbon disclosure can reduce the cost of corporate debt financing and increase the proportion of institutional investors' shareholding, thus improving corporate financial performance. Heterogeneity test results show that the main effect of the enterprise samples in the western and central regions is more significant, indicating that the improvement of the regional economic level will make investors pay more attention to environmental information, but this paper does not carry out an in-depth study of the formation mechanism of this phenomenon, which can be continued to be explored in the subsequent research. The current research on carbon disclosure is still at an early stage, and there are still many aspects that scholars should continue to deepen. First, this paper analyzes corporate carbon disclosure as an indicator as a whole, which can be further refined in subsequent studies to further explore the impact of different aspects of carbon disclosure on corporate financial performance. Second, there are differences in carbon emissions, environmental protection policies, and market competition in different industries, which may lead to differences in the impact of carbon disclosure on financial performance. Future research can select multiple industries for cross-industry comparative studies to reveal the similarities and differences in the relationship between carbon disclosure and financial performance among different industries. Moreover, existing studies focus on the relationship between carbon disclosure and financial performance in the short term, while there are relatively few studies on the long-term impact. In the future, long-term tracking studies can be conducted to reveal the impact of carbon disclosure on the long-term financial performance of enterprises and its dynamic change process. Meanwhile, with the continuous advancement of global climate governance policies, governments are becoming more and more stringent in their requirements for corporate carbon disclosure. There are differences in corporate carbon disclosure behaviors under different environmental policies, and their impacts on financial performance are also different. In addition, companies may adopt strategic disclosure behaviors in response to regulatory requirements when facing policy pressure. How firms implement strategic behaviors such as shaping corporate image and attracting investors and stakeholders through carbon disclosure, and the potential impact of such behaviors on financial performance are also a major focus of future research. Finally, institutional investors, as important stakeholders, may have a direct impact on corporate financial performance through their concerns and reactions to corporate carbon disclosure. In addition to investors, stakeholders such as governments, consumers, and social groups may also have an important impact on corporate carbon disclosure. Future research can explore how these stakeholders influence corporate carbon disclosure behavior and its financial performance.

Based on the above research, this paper puts forward the following suggestions to enterprises: Firstly, enterprises should take into account short-term benefits and long-term development, and put the concepts of low-carbon and environmental protection into every aspect of production and operation. Through technological innovation, management optimization, and other means, they should reduce the carbon emission intensity of enterprises and enhance the efficiency of resource utilization. Secondly, with the increasing concern of society for environmental protection and sustainable development, carbon disclosure has become an important indicator for measuring the green image and market competitiveness of enterprises. Enterprises should abandon traditional concepts and regard carbon disclosure as an indispensable part of their overall development strategy, rather than merely as an additional burden. Third, enterprises located in the western region should make full use of their geographical advantages to demonstrate their environmental responsibility and green business results, which not only helps attract more investors and partners but also enhances their market competitiveness. To further enhance their green competitiveness, enterprises should focus on technological innovation to reduce carbon emissions and improve resource utilization efficiency. Fourth, enterprises should grasp the development opportunities brought by national policy support. Currently, the state has issued relevant laws and regulations to regulate the disclosure of corporate information, and enterprises should strictly abide by the relevant system, actively fulfill their social responsibilities, proactively disclose carbon information, and contribute to the realization of the "dual-carbon" goal.

## Supporting information

**S1 Data.**
(ZIP)

## Author Contributions

**Conceptualization:** Ailing Xu, Yuanyuan Su.

**Data curation:** Yuanyuan Su.

**Funding acquisition:** Ailing Xu.

**Methodology:** Jia Liao.

**Software:** Yingxin Wang, Jia Liao.

**Validation:** Jia Liao.

**Writing – original draft:** Ailing Xu, Yuanyuan Su.

**Writing – review & editing:** Ailing Xu, Yuanyuan Su, Yingxin Wang.

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
