## [Decision Letter · Decision Letter 0]

13 Sep 2024

PONE-D-24-32603

Carbon information disclosure and corporate financial performance— Empirical evidence based on heavily polluting industries in China

PLOS ONE

Dear Dr. Su,

Thank you for submitting your manuscript to PLOS ONE. After careful consideration, we feel that it has merit but does not fully meet PLOS ONE’s publication criteria as it currently stands. Therefore, we invite you to submit a revised version of the manuscript that addresses the points raised during the review process.

**ACADEMIC EDITOR: **Dear authors,

I am really happy to read your paper.

I have received input from three expert reviewers. You should significantly improve the introduction of your manuscript. In addition, the paper needs to be reanalyzed to include the data till 2023. Along with my comments, all reviewers have done a great job in pointing out the directions you should follow to improve the paper, so please attentively read and follow their suggestions.

Additional comments:

1. The tables needs to be organized well.

2. The language of the paper needs to be revised as there are currently a lot of mistakes.

3. We note that one reviewer has recommended that you cite specific previously published works. As always, we recommend that you review and evaluate the requested works to determine whether they are relevant and should be cited. It is not a requirement to cite these works. We appreciate your attention to this request.

4. Please include the data until 2023.

We look forward to receiving your revised manuscript.

Kind regards,

Saddam A. Hazaea, Postdoctoral

Academic Editor

PLOS ONE

2.  Thank you for stating the following financial disclosure: [Fujian Provincial Social Science Foundation Project，grant numbers: FJ2022B082.]. At this time, please address the following queries: a) Please clarify the sources of funding (financial or material support) for your study. List the grants or organizations that supported your study, including funding received from your institution. b) State what role the funders took in the study. If the funders had no role in your study, please state: “The funders had no role in study design, data collection and analysis, decision to publish, or preparation of the manuscript.” c) If any authors received a salary from any of your funders, please state which authors and which funders. d) If you did not receive any funding for this study, please state: “The authors received no specific funding for this work.” Please include your amended statements within your cover letter; we will change the online submission form on your behalf.

3. We note that your Data Availability Statement is currently as follows: [All relevant data are within the manuscript and its Supporting Information files.] Please confirm at this time whether or not your submission contains all raw data required to replicate the results of your study. Authors must share the “minimal data set” for their submission. PLOS defines the minimal data set to consist of the data required to replicate all study findings reported in the article, as well as related metadata and methods (https://journals.plos.org/plosone/s/data-availability#loc-minimal-data-set-definition). For example, authors should submit the following data: - The values behind the means, standard deviations and other measures reported; - The values used to build graphs; - The points extracted from images for analysis. Authors do not need to submit their entire data set if only a portion of the data was used in the reported study. If your submission does not contain these data, please either upload them as Supporting Information files or deposit them to a stable, public repository and provide us with the relevant URLs, DOIs, or accession numbers. For a list of recommended repositories, please see https://journals.plos.org/plosone/s/recommended-repositories. If there are ethical or legal restrictions on sharing a de-identified data set, please explain them in detail (e.g., data contain potentially sensitive information, data are owned by a third-party organization, etc.) and who has imposed them (e.g., an ethics committee). Please also provide contact information for a data access committee, ethics committee, or other institutional body to which data requests may be sent. If data are owned by a third party, please indicate how others may request data access.

Additional Editor Comments:

Dear authors,

I am really happy to read your paper.

I have received input from three expert reviewers. You should significantly improve the introduction of your manuscript. In addition, the paper needs to be reanalyzed to include the data till 2023. Along with my comments, all reviewers have done a great job in pointing out the directions you should follow to improve the paper, so please attentively read and follow their suggestions.

Additional comments:

1. The tables needs to be organized well.

2. The language of the paper needs to be revised as there are currently a lot of mistakes.

3. We note that one reviewer has recommended that you cite specific previously published works. As always, we recommend that you review and evaluate the requested works to determine whether they are relevant and should be cited. It is not a requirement to cite these works. We appreciate your attention to this request.

4. Please include the data until 2023.

Reviewers' comments:

Reviewer's Responses to Questions

**Comments to the Author**

1. Is the manuscript technically sound, and do the data support the conclusions?

Reviewer #1: Yes

Reviewer #2: Partly

Reviewer #3: Partly

2. Has the statistical analysis been performed appropriately and rigorously? 

Reviewer #1: Yes

Reviewer #2: Yes

Reviewer #3: Yes

3. Have the authors made all data underlying the findings in their manuscript fully available?

Reviewer #1: Yes

Reviewer #2: Yes

Reviewer #3: Yes

4. Is the manuscript presented in an intelligible fashion and written in standard English?

Reviewer #1: Yes

Reviewer #2: Yes

Reviewer #3: No

5. Review Comments to the Author

Reviewer #1: Very good. With good analysis of Data Result Conclusion and need to improve the completeness of the writing quality each bab. Especially, the data availability relevant and consistent could answer the question of the paper.

Reviewer #2: The paper has prepared and clearly explained the literature research and methods.

Please revised these points:

(1) Revise grammatical/ typo problems. Please edit in English again.

(2) In the abstract. explain the main contribution of this article.

(3) In the introduction, please explain the gaps in previous research and the focus of this research.

(4) In the research conclusion section, please explain the research focus and future prospects of this paper.

(5) The paper is lacking more latest references. (2022-2024)

For example:

Liu, X., Qi, C., Liu, Y., Xia, Y., & Wu, H. (2023). High-Quality Growth in Rural China: Systems-Based Analysis of Digital Entrepreneurial Ecosystems. Journal of Organizational and End User Computing, 35(1), 1-23.

Xie, Y. (2023). Optimization of Enterprise Financial Performance Evaluation System Based on AHP and LSTM Against the Background of Carbon Neutrality. Journal of Organizational and End User Computing, 35(1), 1-14.

……….

Reviewer #3: The paper analyzes the relationship between carbon information disclosure and corporate financial performance based on the information asymmetry theory, signaling theory, and rent-seeking theory. The results show that enhanced corporate carbon disclosure can significantly improve corporate financial performance, and the main effect is realized by reducing the cost of debt financing and increasing the proportion of institutional investors' shareholding. Some comments are as follows.

(1)This is not a good topic, because both carbon information disclosure and corporate financial performance are affected by a same factor: financial condition of firms. So this study not only the problem of endogeneity that the paper has tried to solve, but also the self-selection problem of the sample that is not considered in the paper.

(2) In the Table 2. Description of variables, the size of firms is not considered in the control variables and it is not reasonable.

(3)In the Table 5. Regression results of carbon disclosure and corporate financial performance, the variables of carbon information disclosure and corporate financial performance should not be the same t time point.

(4) The paper is in a mess in the format and there is no significant research meanings.

6. PLOS authors have the option to publish the peer review history of their article (what does this mean?). If published, this will include your full peer review and any attached files.

Reviewer #1: **Yes: **Shabrina Herawati

Reviewer #2: No

Reviewer #3: No

---

## [Author Response · Author response to Decision Letter 0]

6 Oct 2024

Dear Editors and reviews,

Thank you for your letter and for the reviewers’ comments concerning our manuscript entitled “Carbon information disclosure and corporate financial performance—— Empirical evidence based on heavily polluting industries in China”([PONE-D-24-32603] - [EMID：520 b 0 bb8 bd 4a 2015]).We are grateful for the reviewers’ helpful comments, and hope our revision addresses them all. In the marked copies of the manuscript, we have highlighted the changes made to the original version. We have carefully addressed all the issues raised by the reviewers. We hope they are satisfied with our responses and with the new figures or data we have provided. Thank you again for your help.Furthermore, we would like to show the details as follows:

Editor #

Comments 1: The tables needs to be organized well.

Response 1: Thank you for pointing this out. We apologize that the formatting of the tables creating an obstacle for editors and reviewers to read. We present each table in a more appropriate and consistent format to clearly and visually present our empirical results.（Table 1-9）

Comments 2: The language of the paper needs to be revised as there are currently a lot of mistakes. 

Response 2: We apologize for the poor language of our manuscript. We worked on the manuscript for a long time and the repeated additional and removal of sentences and sections obviously led to poor readability. We have now worked on both language and readability. We really hope that the flow and language level have been substantially improved. 

Comments 3: We note that one reviewer has recommended that you cite specific previously published works. As always, we recommend that you review and evaluate the requested works to determine whether they are relevant and should be cited. It is not a requirement to cite these works. We appreciate your attention to this request.

Response 3: We deeply appreciate your suggestion.The paper <High-Quality Growth in Rural China: Systems-Based Analysis of Digital Entrepreneurial Ecosystems> describes the impact of digital entrepreneurial ecosystems on economic growth in rural China. <Optimization of Enterprise Financial Performance Evaluation System Based on AHP and LSTM Against the Background of Carbon Neutrality>, on the other hand, adopts a comprehensive evaluation method to study the energy companies' financial performance using a comprehensive evaluation method. After carefully reading the two papers recommended by the reviewers, we concluded that they deviated from the topic of this paper and therefore did not include them in the scope of reference. In addition, we referenced more recent literature as requested by the reviewers. （ref 21-24,35-36,39,43-44,56-57）

Comments 4: Please include the data until 2023.

Response 4: We are grateful for the suggestion. We have updated the data to 2023 and the conclusions are consistent with the previous ones.

Reviewer 1 #

Comments 1: Very good. With good analysis of Data Result Conclusion and need to improve the completeness of the writing quality each section. Especially, the data availability relevant and consistent could answer the question of the paper.

Response 1: Thank you for your advice. We tried our best to improve the manuscript and made some changes to the manuscript.We hope the revised manuscript could be acceptable for you.

Reviewer 2 #

Comments 1: Revise grammatical/ typo problems. Please edit in English again.

Response 1: Thank you for pointing this out. We are very sorry for the grammatical error in the article and we have re-edited the article.

Comments 2: In the abstract. explain the main contribution of this article.

Response 2: Thank you for your comments. We have added the contribution of this paper in the abstract. Based on the existing studies, this paper expands the research on the relationship between carbon disclosure and financial performance, and explores the roles played by creditors, institutional investors and other stakeholders in carbon disclosure and corporate financial performance based on signaling theory, rent-seeking theory and sustainable development theory, and further constructs an intermediation model to analyze how carbon disclosure affects corporate financial performance. In addition, considering the existence of development imbalance among regions in China, this paper distinguishes the full sample into three sub-samples for regression separately according to the regions where the heavy polluters are located. This heterogeneity test incorporates inter-regional development differences into the analysis, which provides a reference for China's future implementation of carbon disclosure policies, promotion of the carbon trading market, and achievement of carbon emission reduction targets. （Page 2-3)

Comments 3: In the introduction, please explain the gaps in previous research and the focus of this research.

Response 3: We deeply appreciate your suggestion.We have added to the introduction accordingly. Through the reading of previous literature, we find that the research on the relationship between carbon disclosure and financial performance is deficient in various aspects. Firstly, previous studies have mostly focused on the direct association between carbon disclosure and financial performance, but less on the underlying mechanisms and paths. This shallow research limits our comprehensive understanding of the relationship between the two. And some of the studies may only analyze enterprises in a specific industry or region, lacking extensive cross-industry and cross-region studies, which may lead to insufficient generalizability of the research conclusions. Second, the current theoretical framework on the relationship between carbon disclosure and financial performance is not yet perfect, lacking a unified theoretical foundation and explanatory mechanism. Some studies may be oversimplified in constructing the model and fail to fully consider other factors that may affect the relationship, such as enterprise size, industry characteristics, and policy environment, leading to limited explanatory power of the model. Finally, due to the limitations of factors such as the difficulty and cost of data acquisition, some studies may not be able to obtain a sufficient number of samples or the samples may be under-representative, resulting in possible bias in the conclusions of the studies.In contrast, this paper examines the role played by carbon disclosure on financial performance from the perspectives of institutional investor ownership and the cost of debt financing. To ensure the robustness of the results, this paper refers to previous studies to alleviate the endogeneity problem and the sample self-selection problem. Based on this, the paper further distinguishes the sample into three sub-samples to investigate the impact of carbon disclosure on financial performance in different regions.（Page 5-6）

Comments 4: In the research conclusion section, please explain the research focus and future prospects of this research.

Response 4: Thanks for the suggestion and sorry we missed it. This paper focuses on the two dimensions of institutional investors' shareholding ratio and debt financing cost and analyzes how carbon disclosure affects firms' financial performance through these two critical paths in a China-specific context, using a sample of Chinese A-share listed firms in Shanghai and Shenzhen from 2013 to 2023 in the heavy polluting industry. Besides, this paper argues that the research on the relationship between carbon disclosure and financial performance can be deepened in several directions in the future. First, the evaluation indexes of the degree of carbon disclosure can be further refined to evaluate the carbon disclosure of enterprises from different perspectives. Second, the industries involved in the empirical study will be more diversified, and attention will be paid to the long term impact of carbon disclosure on the financial performance of enterprises, in order to reveal the differences in the relationship between different industries and in the long term perspective. Meanwhile, the interaction between the policy environment and corporate behavior will become the focus of the study to explore the policy-driven corporate carbon disclosure behavior and its impact on financial performance. Finally, the expansion of the stakeholder perspective will cover the responses of investor behavior, government, consumers and other parties to corporate carbon disclosure and its potential impact on financial performance.（Page 29-30）

Comments 5: The paper is lacking more latest references. (2022-2024)

For example:

Liu, X., Qi, C., Liu, Y., Xia, Y., & Wu, H. (2023). High-Quality Growth in Rural China: Systems-Based Analysis of Digital Entrepreneurial Ecosystems. Journal of Organizational and End User Computing, 35(1), 1-23.

Xie, Y. (2023). Optimization of Enterprise Financial Performance Evaluation System Based on AHP and LSTM Against the Background of Carbon Neutrality. Journal of Organizational and End User Computing, 35(1), 1-14.

Response 5: We sincerely appreciate the valuable comments. We deeply appreciate your suggestion.The paper <High-Quality Growth in Rural China: Systems-Based Analysis of Digital Entrepreneurial Ecosystems> describes the impact of digital entrepreneurial ecosystems on economic growth in rural China. <Optimization of Enterprise Financial Performance Evaluation System Based on AHP and LSTM Against the Background of Carbon Neutrality>, on the other hand, adopts a comprehensive evaluation method to study the energy companies' financial performance using a comprehensive evaluation method.After reading the two papers you recommended, we concluded that they deviated from the topic of this paper and therefore did not include them in the scope of reference. In addition, we referenced more recent literature.（ref 21-24,35-36,39,43-44,56-57）

Comments 6: For 1 ref too much sentence.Please summarize it.

Response 6: Thank you for pointing this out. We've restated and streamlined that sentence. In the context of global commitment to promoting carbon emission reduction, corporate carbon disclosure has gradually become a hot topic of academic discussion. Corporate carbon disclosure is an important way for stakeholders to understand the environmental dynamics of enterprises. However, in the primary stage of carbon disclosure, enterprises face the trade-off between environmental and economic benefits. Against this background, this paper explores the relationship between carbon disclosure and financial performance in an attempt to explore whether there is a win-win situation between corporate environmental performance and financial performance.（Line 60）

Comments 7: Line 58. Maybe there are other articles to compare.

Response 7: We deeply appreciate your suggestion. To learn more about the current state of carbon disclosure, we looked for additional literature and compared it with other countries. 

 Compared with other countries and regions, the European Union and the United Kingdom have more mature and perfect systems in the field of carbon information disclosure. Specifically, the carbon information disclosure requirements implemented by the EU constitute a benchmark in this field, which stipulates that firms should include strategic considerations, policy frameworks, and evaluation indicators of the effectiveness of specific actions in the relevant disclosure content. In addition, the U.S. Securities and Exchange Commission proposed new climate change disclosure rules in 2022, requiring relevant companies to disclose greenhouse gas emissions information in accordance with international standards and provide corresponding data for certification.Scholars have studied the willingness of developed and developing countries to make voluntary carbon disclosures. The study found that developing countries have a greater demand for carbon information compared to developed countries. When enterprises disclose more carbon information, their value will be enhanced. （Page 4-5）

Comments 8: Line 77.Please add theories related to the environment.

Response 8: We think your suggestion to add theories related to the environment is important.In view of your suggestion, we have added the sustainability theory to the article to support our research. The theory of sustainable development points out that enterprises should protect the ecological environment and change the mode of development while developing, and should not oppose environmental protection to economic development. Under the framework of the theory of sustainable development, enterprises send positive signals to the market through carbon information disclosure that they are committed to environmental protection and fulfill their social responsibilities, which helps them build up a green image and attract more investors and partners who are concerned about sustainable development. By cooperating with investors who share the same goals, enterprises can send positive signals to the outside world, increase the proportion of institutional investors' shareholding, reduce the cost of debt financing, and thus improve the financial performance of enterprises.（Page 10-11）

Comments 9: Line 117.Make sure including from other countries. 

Response 9: Thank you for your comment. We read more literature to understand the progress of research on carbon disclosure in different countries.Bedi et al. based on a study of 100 Indian firms from 2018-2021 found that carbon disclosure by firms can accelerate the process of climate disclosure by global firms, as well as enhance the financial performance of firms. Using a sample of Korean firms, Lee et al. found that voluntary carbon disclosure by firms can reduce the risk of share price crash and thus gain in the capital market.Sun et al. noted that improving the quality of carbon disclosure can effectively reduce the information asymmetry between firms and stakeholders, thus enhancing corporate financial performance.Nazila and Fauziah, based on the data of 27 firms in Indonesia for four years Nazila and Fauziah, based on four-year data from 27 firms in Indonesia, found that firms with more comprehensive disclosure of carbon emissions and social responsibility information have higher levels of financial performance. We found that scholars from different countries are largely consistent in their interpretations and findings on the relationship between carbon disclosure and financial performance.（Line 190）

Reference:

1.Bedi A, Singh B. Exploring the impact of carbon emission disclosure on firm financial performance: moderating role of firm size. Management Research Review. 2024.

2.Lee G, Bae M, Sohn J, Han C, Cho J. Does voluntary environmental information disclosure prevent stock price crash risk? – Comparative analysis of chaebol and non-chaebol in Korea. Energy economics. 2024;131:107394–4.

3.Sun ZY, Wang SN, Li D. The impacts of carbon emissions and voluntary carbon disclosure on firm value. Environ Sci Pollut Res. 2022; 29:60189–60197.

 4.Nazila Nazwa, Fauziah Aida Fitri. Can Carbon Emission Disclosure, Environmental Performance, and Corporate Social Responsibility Improve Firm Value in Indonesia? 2022 International Conference on Decision Aid Sciences and Applications (DASA). 2022: 1163-1167.

Comments 10: Line 510.Give the reason why.

Response 10: We are grateful for the suggestion. We apologize for not explaining clearly why this regional difference occurs. We believe there are several reasons for this phenomenon, including the following. First, the economic development of the central and western regions is relatively lagging behind, and enterprises may still be in the initial stage in terms of environmental awareness and carbon reduction technology. Therefore, when enterprises begin to pay attention to and disclose carbon information, this behavior can enhance their sense of social responsibility and reduce information asymmetry, which is easier to obtain the recognition of stakeholders, and then have a positive impact on financial performance. Second, in the central and western regions, where heavy pollution industries are more developed, the government will incentivize and guide enterprises to obtain more tax incentives and government subsidies by disclosing

---

## [Decision Letter · Decision Letter 1]

29 Oct 2024

Carbon information disclosure and corporate financial performance— Empirical evidence based on heavily polluting industries in China

PONE-D-24-32603R1

Dear Dr. Jia Liao

We’re pleased to inform you that your manuscript has been judged scientifically suitable for publication and will be formally accepted for publication once it meets all outstanding technical requirements.

Kind regards,

Saddam A. Hazaea, Postdoctoral

Academic Editor

PLOS ONE

Additional Editor Comments (optional):

Dear authors,

Please note that tables 2 and 4 are still not cited in the text and need to be revised before final acceptance. In addition, you have also mentioned table ...... on page 15

(The relevant variables are shown in the following table. My question is which table ).

On page 17 you mention the table two times as (Which table)???????

(According to the table, the correlation coefficient between Roa and Cdi is 0.086, which is significant at the 1% level, which indicates that corporate carbon disclosure has a significant positive impact on financial performance in the whole sample, and this finding is the same as the existing literature [19,21-25]. In addition, the absolute value of the correlation coefficients of the variables is less than 0.5, which indicates that there is no serious multicollinearity relationship among the variables. For the variables whose absolute values of the correlation coefficients in the table )

You need to check this problem carefully.

All the best.

Reviewers' comments:

Reviewer's Responses to Questions

**Comments to the Author**

1. If the authors have adequately addressed your comments raised in a previous round of review and you feel that this manuscript is now acceptable for publication, you may indicate that here to bypass the “Comments to the Author” section, enter your conflict of interest statement in the “Confidential to Editor” section, and submit your "Accept" recommendation.

Reviewer #1: All comments have been addressed

Reviewer #2: All comments have been addressed

2. Is the manuscript technically sound, and do the data support the conclusions?

Reviewer #1: Yes

Reviewer #2: (No Response)

3. Has the statistical analysis been performed appropriately and rigorously? 

Reviewer #1: Yes

Reviewer #2: Yes

4. Have the authors made all data underlying the findings in their manuscript fully available?

Reviewer #1: Yes

Reviewer #2: Yes

5. Is the manuscript presented in an intelligible fashion and written in standard English?

Reviewer #1: Yes

Reviewer #2: Yes

6. Review Comments to the Author

Reviewer #1: The authors improve well, with optimistic to the better article output, they did revise well I think

Reviewer #2: The manuscript was clearly written and the basic ideas can be easily followed.

The paper has prepared and clearly explained the literature research and methods.

This article has been completely revised.

I suggest the paper should be accepted.

7. PLOS authors have the option to publish the peer review history of their article (what does this mean?). If published, this will include your full peer review and any attached files.

Reviewer #1: **Yes: **Shabrina Herawati

Reviewer #2: No

---

## [Editor Report · Acceptance letter]

7 Jan 2025

PONE-D-24-32603R1 

PLOS ONE

Dear Dr. Liao, 

I'm pleased to inform you that your manuscript has been deemed suitable for publication in PLOS ONE. Congratulations! Your manuscript is now being handed over to our production team.

Kind regards, 

on behalf of

Dr. Saddam A. Hazaea 

Academic Editor

PLOS ONE